# The Impact of Environmental Factors on the Development of Autoimmune Thyroiditis—Review

**DOI:** 10.3390/biomedicines12081788

**Published:** 2024-08-07

**Authors:** Wojciech Cyna, Aleksandra Wojciechowska, Weronika Szybiak-Skora, Katarzyna Lacka

**Affiliations:** 1Student’s Scientific Society, Endocrinology Section at the Department of Endocrinology, Metabolism and Internal Medicine, Poznan University of Medical Sciences, 60-355 Poznan, Poland; cynawojtek2@gmail.com (W.C.); alleks.woj@gmail.com (A.W.); weronikaszybiak@gmail.com (W.S.-S.); 2Department of Endocrinology, Metabolism and Internal Medicine, Poznan University of Medical Sciences, 60-355 Poznan, Poland

**Keywords:** autoimmune thyroiditis, environmental factors, iodine, vitamin D, selenium, viruses, COVID-19, microbiota

## Abstract

Autoimmune thyroiditis (Hashimoto’s thyroiditis) is the most common autoimmune disease. It most often manifests itself as hypothyroidism but may also present with euthyroidism or even hyperthyroidism. The etiopathogenesis of autoimmune thyroiditis is still unclear. However, in addition to genetic and epigenetic factors, many environmental factors are known to increase the risk of developing AIT. In this review, we aimed to collect and analyze data connected with environmental factors and autoimmune thyroiditis development. Our review indicates iodine intake, vitamin D deficiency, selenium deficiency, viral infections caused by Epstein–Barr Virus (EBV), Human parvovirus B19 (PVB19), Human herpesvirus 6A (HHV-6A) and Severe acute respiratory syndrome coronavirus 2 (SARS-CoV-2), bacterial infection caused by Helicobacter pylori, microbiome disruption, medications such as interferon-alpha and tyrosine kinase inhibitors, as well as stress, climate, and smoking can influence the risk of the occurrence of autoimmune thyroiditis. Having knowledge of risk factors allows for making changes to one’s diet and lifestyle that will reduce the risk of developing the disease and alleviate the course of autoimmune thyroiditis.

## 1. Introduction

Autoimmune thyroiditis (AIT), also known as Hashimoto’s disease, is the most common autoimmune disorder which most often manifests itself as hypothyroidism but may also present with euthyroidism or even hyperthyroidism.

The first mention of autoimmune thyroiditis dates back to 1912, when a Japanese doctor, Hakaru Hashimoto, described four cases of patients with thyroid lesions in the form of goiter [1].

As described by Hashimoto, AIT is clinically characterized by a diffused enlargement of the thyroid gland, known as goiter [1]. During palpation, an enlarged thyroid gland is usually painless and has a hard and smooth surface. However, as the condition develops, the thyroid gland becomes smaller and nodules may begin to appear [2].

The incidence rate of AIT depends on gender, age, and the individual traits of a given population. The frequency of this disorder being diagnosed is on a trend of growth, and it is estimated that approximately 5% of Caucasians are affected by it [2]. A significantly larger proportion of AIT cases are women, which are estimated to be at least 2% of the total female population [3]. Additionally, in the female population, Hashimoto’s disease was found to be more common among patients with Polycystic Ovary Syndrome (PCOS) [4,5]. The incidence rate of AIT increases with age, with a peak between the ages of 45 and 65; however, it can also be observed in children [2,3]. Moreover, particularly in the pediatric group, a higher incidence rate of this disease is diagnosed in the context of specific chromosome aberrations, such as Down syndrome, Turner syndrome, or Klinefelter syndrome [6].

According to the American Thyroid Association (for 2023) [7], the diagnostic criteria for autoimmune thyroiditis include

symptoms of hypothyroidism combined with elevated Thyroid Stimulating Hormone (TSH) levels, with or without low thyroid hormone levels;enlargement of the thyroid gland (goiter);elevated thyroid antibody levels.

In Hashimoto thyroiditis (HT), the prevailing diagnostic antibodies are thyroid peroxidase antibodies (TPOAb) and thyroglobulin antibodies (TgAb), with the dominance of TPOAb, which occurs in up to 95% of patients [8,9,10]. Antibodies against thyroxine (T4), triiodothyronine (T3), pendrin (PDN), or sodium iodide symporter (NIS) can rarely be found in AIT [7,11]. Apparent hypothyroidism is defined as a high level of TSH and a low free thyroxine (fT4) level [10]. Laboratory hypothyroidism occurs with high TSH and normal fT4 and free triiodothyronine (fT3) levels [10]. Other diagnostic methods in approaching HT include ultrasonography (USG) of the thyroid gland and fine needle aspiration (FNA) [9,12]. The most commonly observed changes in the USG are enlargement, hypoechogenicity of the thyroid parenchyma, and hypervascularity, although it depends on the phase of HT [9,10,13]. Single or multiple nodules in the parenchyma of the thyroid may be present in the nodular type of HT [9]. Cytology findings performed by FNA most often present a lymphocytic thyroiditis with lymphoid follicles and metaplasia of oncocytic cells (Figure 1) [9,14].

Hashimoto’s disease and the endocrine disorders that occur in its course can affect the functioning of most systems and organs and induce many health problems.

In a group of AIT patients, a significantly higher incidence of depression and anxiety disorders was observed compared to the healthy control group [15]. Men with hypothyroidism and also in the course of AIT are more likely to experience erectile dysfunction, which may also be associated with an increased risk of depression. Sexual dysfunction in men is regulated by thyroid hormone replacement therapy. However, the relationship between female sexual dysfunction and thyroid disease remains unclear [16].

The presence of AIT may affect not only the sexual health of women and men but also the effectiveness of procreation. A study by Quintino-Moro et al. showed infertility, defined as the absence of pregnancy in sexually active women having regular unprotected intercourse with a male partner for an exposure period of at least 12 months, in 47% of women with Hashimoto’s disease [17].

The level of thyroid hormones also affects the cardiovascular system. Hypothyroidism disturbs the cardiac output and systemic vascular resistance through impaired vascular smooth muscle relaxation and decreased levels of endothelial nitric oxide. Moreover, impaired vascular smooth muscle relaxation promotes atherosclerosis and increases the risk of ischemic heart disease [18].

Patients with AIT may also be at an increased risk of developing not only thyroid cancer but also breast cancer, lung cancer, gastrointestinal cancer, genitourinary cancer, blood cancer, and prolactinoma [19].

The etiopathogenesis of autoimmune thyroiditis is still unclear. However, it is known that it is associated with a complex interaction between multiple predisposing genes, epigenetic factors, as well as environmental triggers. The impact of AITD on a huge number of health disorders indicates the need for effective treatment and the avoidance of environmental risks, which can influence the development and course of AIT.

## 2. Materials and Methods

The aim of our review was to gather and summarize the knowledge about the impact of specific environmental factors on AIT.

The authors searched the PubMed, Scopus, Google Scholar, and Cochrane Library databases. The articles were searched based on the keywords found in the title or abstract (only in the title using Google Scholar): “Hashimoto” and “specific factor”. Automatic search filters were used to find articles only with an available free full text published after 1970. The inclusion criteria mentioned only original articles related to the topic of this review after title/abstract screening, publications in the English language, and free full access to the article. All articles that did not meet the inclusion criteria were excluded.

## 3. Environmental Factors in Autoimmune Thyroids Disorders

### 3.1. Iodine Excess

Iodine is the micronutrient necessary to produce thyroid hormones, which affect growth in childhood, metabolism, and other physiologic processes of the body [20,21]. The thyroid gland stores iodine, oxidized forms of which are attached to thyroxine residues in order to create mono- and diiodothyronine. The process of merging mono- and diiodothyronine is fundamental to producing T4 and T3, final forms of thyroid hormones. The most important dietary sources of iodine are milk, eggs, and meat. Geographic regions play a vital role in iodine daily intake considering the diet and culture [21]. 

In the past, iodine deficiency was a vast problem causing overt hypothyroidism in a great number of people, especially in regions with low iodine nutritional statuses [22,23]. Food iodization, mainly salt, to maintain proper iodine levels in the population [22] resulted in a decrease in iodine deficiency; however, the amount of AIT has noticeably risen. Additionally, countries with remaining iodine deficiency had less AIT cases than countries with higher levels of iodine in their diet [23]. 

An increased titer of TPOAb and TgAb is observed in patients with AIT. It is considered that excessive iodine intake is responsible for Tg iodination, which results in the high immunogenicity of this protein [23]. Additionally, the inflammatory status in patients with AIT may be triggered by cytokines production, especially interferon-γ (IFN-γ). The inflammatory reaction is induced by MHC class II, affecting the activation of macrophages and intercellular adhesion molecule-1 (ICAM-1), which results in the recruitment migration of leukocytes to the area of inflammation [23]. Highly immunogenic Tg causes an increase in IFN-γ inducible protein 10 (IP-10), which is considered to correlate with the severity of AIT [23] (Figure 2).

A study on the Ukrainian population showed an increased titer of TgAb and TPOAb depending on the iodine concentration in specific water supplies [24]. Shanxii province in China was one of the most iodine-deficient areas, in which the worldwide salt iodization program resulted in a higher incidence of AIT, hyperthyroidism, and thyroid cancer [22]. Interestingly, a similar study was performed in Slovenia, which found an association between a higher iodine intake and a higher fT4/fT3 ratio [25]. Hongyan et al. confirmed that elevated levels of urinary iodine (UI) and of autoantibodies (TPOAb and TgAb) positively correlate with the severity of AIT. This phenomenon corresponds with the cytotoxic effect of auto-Abs in the pathology of AIT [26]. Additionally, a correlation between a high iodine nutritional status and AIT has been found in younger patients (with a mean age of 9) [27]. Children with elevated urinary iodine (UI) are found to have higher TSH and lower fT4 and fT3 levels [20]. Moreover, regarding the effect of iodine on pregnancy, a correlation was found between the usage of povidone iodine disinfection during the delivery and transitory hypothyroidism with an exclusion of congenital hypothyroidism at 1 year of age in the large birth cohort group in Japan [28]. All of the mentioned articles are summarized in Table 1.

### 3.2. Vitamin D Deficiency

Vitamin D is known for its effect on maintaining calcium–phosphorus homeostasis and bone metabolism [29,30]. The immunoregulatory effect of vitamin D involves inhibiting the production of pro-inflammatory cytokines containing interleukins (IL) 1, 6, 8, 12 and tumor necrosis factor (TNF-α) and reducing the differentiation of dendritic cells through a decreased number of MHC class II proteins [31]. Additionally, 1,25(OH)2D affects B and T lymphocytes, causing a decrease in the production of antibodies and a decrease in the production of IL-17 [32]. Due to this, vitamin D is considered an anti-inflammatory factor [33]. The influence of this vitamin on several endocrine and autoimmune diseases has been confirmed, e.g., rheumatoid arthritis, type I diabetes mellitus, multiple sclerosis, and psoriasis [30,32,34]. The supplementation of vitamin D is important in reducing the inflammation processes (Figure 3), especially in obese people, who are mostly deficient in vitamin D [35].

The effect of vitamin D on AIT is still being widely researched. Many articles confirm vitamin D deficiency in patients with diagnosed HT. However, whether this is a risk factor or simply a correlation in AIT is unclear. Lower levels of 25(OH)D3 can be found in most patients with AIT. Considering the immunoregulatory effect of vitamin D, these results were expected. Vitamin D deficiency is common in patients with diagnosed HT, especially in geographic regions with low levels of sunlight throughout the year, like Poland, which was proven in a study conducted by Maciejewski et al. [34]. Moreover, lower levels of vitamin D in patients with AIT correlate with higher titers of pro-inflammatory cytokines like IL-1B, IL-8, and TNF-α [33]. Interestingly, a correlation between cognitive impairment and AIT was discovered—HT patients with lower vitamin D levels received lower MoCA (Montreal Cognitive Assessment) scores [36]. AIT patients are more likely to have Vitamin D deficiency, whose titers correlate negatively with the TSH level [37,38]. Regarding younger patients, it was observed that children with AIT, similar to adults, have significantly lower vitamin D titers, found especially in children with profound hypothyroidism [39,40].

On the contrary, there are articles and studies that did not confirm the correlation between low serum vitamin D levels and AIT [41,42,43,44,45,46]. However, other interesting connections were found. Vitamin D levels were connected with the level of thyroxin [47], and low concentrations of vitamin D in AIT patients may be associated with the severity of HT [44]. Every mentioned study is summarized in Table 2.

### 3.3. Selenium Deficiency

Selenium is an important micronutrient known for its antioxidant properties, protecting the body against oxidative stress and preventing inflammatory, neurological and cardiovascular diseases [49,50]. The thyroid gland is an organ with one of the highest levels of selenium, containing a large number of selenoproteins, which indicates the great importance of selenium in the synthesis, activation, and metabolism of thyroid hormones [49]. This element also plays a vital part in the proper functioning of the immune system. These properties are of particular significance in the pathogenesis of autoimmune thyroiditis, the development of which may be predisposed by chronic selenium deficiency.

The level of selenium in the body is significantly influenced by the amount of this trace element in the soil in the inhabited area, as well as the supply of selenium in the diet, which was demonstrated in a study conducted by Qian Wu et al., in which selenium deficiency was almost twice as common in people living in an area with a lower selenium intake compared to a population living in an area with an adequate selenium intake. Moreover, the number of new cases of AIT that were diagnosed in the area with the lower selenium intake was three times higher in comparison with that in the area with an adequate selenium intake, which indicates a significant influence of selenium deficiency on the pathogenesis of AIT [51].

Similar results were observed in another study that focused on the role of selenium, as well as the variety of selenoproteins and oxidative stress, in the pathogenesis of AIT [52]. This study showed that in the group of AIT patients, the level of selenium was significantly lower, while, among other things, the level of selenoprotein H (SelH) was significantly higher compared to that in the group of healthy people. SelH, like other selenoproteins, affects the biosynthesis and metabolism of thyroid hormones. Additionally, this particular selenoprotein is characterized by its peroxidase activity and DNA binding function, which allow it to counteract oxidative stress in the cell nucleus. An increased level of SelH in AIT patients may indicate an enhanced antioxidant response to the increased oxidative stress associated with Hashimoto’s thyroiditis [52].

Increased oxidative reactions, as well as lower Se levels, were also detected in patients with AIT in a study conducted by Rahim Rostami et al. [53]. The outcomes additionally showed that in AIT patients, the concentrations of TSH, thyroglobulin, antibodies against thyroid peroxidase, and iodine in urine were higher compared to those in the control group, which emphasizes the importance of concomitant selenium deficiency and increased iodine levels in the progression of thyroid dysfunction [53].

Studies have also been conducted that have shown a positive effect of selenium supplementation on thyroid function in AIT through its reduction in the titer of thyroid autoantibodies (TPOAb, TgAb), its lowering of TSH, and its increase in antioxidant activity and Treg levels, which may indicate a reduction in autoimmunity [54,55] (Figure 4). Some studies have also shown the possibility of restoring euthyroidism in subclinical hypothyroid patients with AIT through selenium supplementation [56,57]. Moreover, in the study by Pirola et al., the restoration of euthyroidism was found in approximately 10 times more participants in the case group than in the control group (which was not treated with selenium) [56]. All of the mentioned articles are summarized in Table 3.

### 3.4. Viral Infections

Another environmental factor that may have a potential impact on the development of autoimmune thyroiditis is viral infections. It is assumed that infections caused by viruses may be involved in triggering AIT thanks to their molecular mimicry, which enhances autoimmune responses [58,59] (Figure 5). Many studies have been carried out to check the relationship between viral infection and the further development of Hashimoto’s disease, but only a few of them were carried out on a large group of people, and the results were not clear in all of them, which indicates the need to conduct further research. Of all the viruses potentially linked to the development of AIT, Epstein–Barr Virus (EBV), human parvovirus B19 (PVB19), human herpesvirus 6A (HHV-6A), and severe acute respiratory syndrome coronavirus 2 (SARS-CoV-2) are the ones considered most likely to influence the development of the condition.

Epstein–Barr Virus is a virus from the herpes family; it is responsible for causing infectious mononucleosis. It also has oncogenic potential, being associated with the pathogenesis of some lymphomas and cancers, such as Burkitt’s lymphoma and nasopharyngeal cancer. 

Research shows that EBV may also influence the pathogenesis of AIT [60,61]. In order to determine this relationship, in addition to measuring thyroid parameters, it is also helpful to examine the serological profile of the EBV virus, which was carried out in the study by Assaad et al. [61]. The results of this study showed that the mean concentrations of EBV early antigen (EA) IgG and EBV viral capsid antigen (VCA) IgG antibodies were significantly higher in the AIT group compared to those in the control group, which may indicate a possible correlation between EBV and AIT [61].

Another viral pathogen that may be associated with the etiopathogenesis of autoimmune thyroiditis is human parvovirus B19. It is one of the smallest DNA viruses belonging to the *Parvoviridae* family. Parvovirus B19 is best known for causing fifth disease, also called erythema infectiosum, which most commonly occurs in children and presents itself with a rash on the cheeks.

A study by Zahra Heidari and Maede Jami showed that the incidence rate of parvovirus B19 infection (determined by elevated levels of IgG antibodies against parvovirus B19) in a group of AIT patients was higher than that in a control group [62]. Additionally, a significant positive correlation was found in the group of patients with AIT between PVB19 IgG and anti-TPOAb and anti-TgAb [62]. These outcomes may indicate a probable connection between PVB19 infection and Hashimoto’s disease; however, more prospective studies are required to confirm this hypothesis. 

According to a study conducted by Noorossadat Seyyedi et al., human herpesvirus 6A is also a pathogen that may have a possible influence on the development of AIT, as its genetic material was detected in serum samples of AIT patients as opposed to healthy euthyroid people [63]. HHV-6A is a double-stranded DNA virus that belongs to the *Herpesviridae*. HHV-6 has been shown to be more common in patients with neuroinflammatory diseases such as multiple sclerosis, and its levels were also increased in patients with Alzheimer’s disease, which would indicate its neurovirulence [58,64]. Furthermore, there are studies indicating the association of HHV-6A with infertility in women [65,66,67]. This shows that this virus can influence the development of many different disorders.

There were also reports on the potential impact of severe acute respiratory syndrome coronavirus 2 on the development of AIT. It still requires many studies conducted over a number of years to confirm this hypothesis. SARS-CoV-2 is an RNA virus belonging to the *coronaviridae* family. It causes an acute respiratory disease, called COVID-19, which is mainly manifested by fever, shortness of breath, dry cough, and shallow breathing. Although, in most cases, the course of the disease is mild. Moreover, we still know too little about the long-term impact of SARS-CoV-2 on human health and about all possible complications of COVID-19 on individual body systems.

In a study by Tesch et al., researchers detected a higher likelihood of acquiring autoimmunity for patients without a preexisting autoimmune disease, as well as a higher likelihood of developing another autoimmune disease for patients with a prior autoimmune disease after SARS-CoV-2 infection compared to controls without COVID-19 in the past. Among the diagnosed autoimmune diseases in patients after SARS-CoV-2 infection, the most common was autoimmune thyroiditis, along with Sjögren syndrome and rheumatoid arthritis. Additionally, it was observed that a more severe course of COVID-19 was linked with a greater risk of a newly diagnosed autoimmune disease [59]. 

Also, enteroviruses were detected in thyroid tissue collected from patients with AITD in a study conducted by Weider et al. In patients with AITD, enterovirus was detected in 51% of cases, moreover, HHV-6 and PBV19 were also found in 30% and 22%, respectively. The species of enteroviruses detected in thyroid belonged mainly to Coxsackievirus A and B, which was confirmed by partial genome sequencing [68]. Some studies also indicate a relationship between enterovirus infection in pregnant women and the development of AITD in their children, but the results were not statistically significant [69]. 

All of the mentioned data are summarized in the Table 4.

In addition, other pathogens are also considered responsible for the pathogenesis of AITD, such as human T-cell lymphotropic virus type 1 (HTLV-1), rubella, and mumps virus. Interestingly, measles–mumps–rubella vaccination has no influence on the autoimmunity process in the thyroid tissue [69]. However, further research is needed to clarify whether or not these viruses are responsible for autoimmune thyroiditis.

### 3.5. Bacterial Infections

Bacteria that may potentially play a part in the etiopathogenesis of Hashimoto’s disease include *Helicobacter pylori*. This pathogen is classified as a Gram-negative bacilli, which inhabits the gastric mucosa in almost half of the human population.

The laboratory parameters used in the diagnosis of *Helicobacter pylori* include the serum concentration of IgG antibodies specific for *Helicobacter pylori* and anti-CagA antibodies, the prevalence of which was significantly higher in patients with AIT compared to the control group [70,71]. Moreover, in a study conducted by Natale Figura et al., an investigation of the structural homology of some thyroid proteins with *Helicobacter pylori* antigens showed that both the studied thyroid proteins and numerous bacterial antigens share common putative conserved domains. These results showed that *Helicobacter pylori* infection, as well as a variant with CagA+ strains, have an important association with AIT, presumably because of the molecular mimicry and increased inflammation [71].

Moreover, metaplasia, atrophy, or dysplasia were found in the biopsy samples of a large number of patients with autoimmune thyroid disease (AIT or Graves’ disease) in the study performed by Maria Pina Dore et al. [72]. 

However, there is also research such as the study by Haim Shmuely et al., which did not show similar conclusions to those of previous studies, thus contradicting the theory about the influence of chronic *Helicobacter pylori* infection on the development of AIT [73].

### 3.6. Microbiome Disruption

The human gut microbiome is a wide colony of microorganisms that inhabit the digestive tract. Bacteria predominate in this group, with the largest number and diversity found in the large intestine. Intestinal bacteria participate in the synthesis of vitamin K, biotin, and some hormones. Additionally, they play a role in the fermentation of dietary fiber, metabolizing xenobiotics, sterols, and bile acids and also stimulating the immune system to eliminate pathogenic microorganisms. The composition of intestinal microflora varies depending on the genes, type of delivery, age, living environment, consumed foods, medications taken (including antibiotics), exposure to stress, as well as co-occurring diseases. The disruption of the gut microbiota composition has been particularly linked to many inflammatory conditions, including autoimmune disorders [74,75]. These changes to the gut microbiome may be a consequence of autoimmune diseases, in addition to being a contributing factor to their development. 

A study by Leonardo César de Freitas Cayres et al. confirmed the importance of dysbacteriosis in influencing the disruption of intestinal microflora and increasing intestinal permeability, which may lead to the entry of bacterial antigens into the bloodstream and, consequently, to the activation of the immune system, which may predispose to the development of AIT [76]. Furthermore, researchers have shown that the modulation of the intestinal microbiota through diet directly affects inflammatory processes due to the production of microbiota metabolites and their effects on immune cells in the intestinal mucosa [76].

Another study, devoted to the analysis of gut microbiota diversity, showed a significant difference in the intestinal microflora of healthy people compared to patients with AIT. *Klebsiella*, *Bilophila*, and *Lachnoclostridium* predominated in the intestinal microbiota of the control group, and *Akkermansia*, *Shuttleia*, *Clostriworthdia*, *Bifidobacterium*, and *Lachnospiraceae* dominated in the gut microbiome of people with AIT [77,78]. Additionally, the results of this study showed that some bacterial colonies are regulated by the hormone of free triiodothyronine (FT3), which suggests that thyroid hormones play vital roles in the regulation of gut microbes as AIT develops [77].

A study by Liu et al. also focused on the alterations in the composition of the intestinal microbiome in AIT patients and showed a lower diversity and abundance of intestinal bacteria in the gut microbiota of AIT patients, with the lowest levels in those with hypothyroidism, compared to controls [79]. However, *Phascolarctobacterium* was found to be more abundant in the microflora of patients with hypothyroidism than in other groups of patients, which may indicate the involvement of this bacteria in the development of AIT [79].

Since these findings showed that dysregulated gut microflora may contribute to the development of AIT, this led some researchers to conduct studies on manipulating the composition of the intestinal microbiota through fecal microbiota transplantation (FMTs). One of the most recent studies is a trial conducted by Aline C Fenneman et al., examining the effect of FMTs on the thyroid reserve in patients with subclinical AIT, with the aim of restoring a normal intestinal microbiome and halting the progression of the disease by weakening autoimmune processes [80]. As this study is still ongoing, we can only hope that it will provide significant evidence that will help uncover distinct patterns within the intestinal microbiome that may be potentially linked to improved thyroid function. Therefore, this may enable the future use of clinical microbial-targeted therapies in people at risk of developing AIT [80].

### 3.7. Medications

One of the environmental factors that may trigger and affect AIT is the usage of specific drugs. In this systematic review, interferon-α, amiodarone, lithium, and tyrosine kinase inhibitors (TKIs) were taken into consideration

#### 3.7.1. Interferon-Alpha

Interferon-α (IFN-α) is a cytokine produced by the host organism, affected by the viral infection [81]. It plays a key role in the immunologic response to the infection by way of stimulating the chemokine and cytokines of a proinflammatory function, as well as elevating the cytotoxicity of lymphocytic T cells [81]. IFN-α regulates cell apoptosis, growth, and resistance to viral infections. This cytokine is used in the therapy of hepatitis B virus (HBV) and hepatitis C virus (HCV) [82,83]. The classic therapy consisting of IFN-α and ribavirin in HCV is associated with many side effects of this treatment. One of the side effects may be thyroid dysfunctions [84]. Some potential mechanisms of such effect are being considered. Faustino et al. [81] presents three paths in which IFN-α may stimulate autoimmunity in thyroid diseases. According to this study, IFN-α stimulates Tg production and destruction via the lysosomal-dependent system at the same time. Such phenomenon may lead to the release of potentially immunogenic Tg peptides and stimulate the autoimmunologic pathomechanism of thyroid diseases [81]. Moreover, the authors suggested the impact of IFN-α on endoplasmic reticulum (ER) stress in thyrocytes, which may lead to the apoptosis of thyroid cells [81]. Additionally, the increased exposure of thyroid cells to IFN-α supposedly triggers autophagic flux, which has an impact on the pathogenesis of autoimmune diseases; however, the role of this mechanism remains unclear [81].

A correlation between IBT (interferon- α based therapy) and the occurrence of AIT in the course of HCV and HBV treatment was observed [82,83]. The usage of pegylated IFN- α and ribavirin (RBV) results in a higher incidence rate of thyroid disfunction (TD), especially in female patients and those with a history of goiter or hyperlipidemia compared to the controls [85]. Interestingly, the authors noticed that abnormal levels of TPO-Ab in patients were correlated with a better effect of the IFN-α treatment in HBV [83]. Moreover, the usage of DAA (direct-acting antivirals) has limited or no influence on TD, including AIT in comparison to the IFN-α-based treatment of HCV infection [85].

#### 3.7.2. Amiodarone

Amiodarone may affect the thyroid in two possible ways. First, there is the thyroid cytotoxicity due to the iodide content in this medication [86,87]. Another way of amiodarone influencing thyroid function is through its inner inhibitive effect on deiodinases [86]. Deiodinase is an enzyme necessary for converting thyroid hormones to different forms—T4, T3, reverse T3 (rT3), and T2. Deiodinase I (D1) and deiodinase II (D2) convert T4 into T3, and D2 additionally controls the intracellular level of T3 [86]. Amiodarone affects thyroid hormone regulation through the inhibition of D1 and D2, resulting in a lower concentration of the biologically active form of thyroid hormones—triiodothyronine [86]. However, amiodarone’s effect on autoimmunity is controversial. To this day, very few studies have been conducted in order to consider the potential impact of amiodarone on AIT. 

According to the studies, the correlation between the usage of amiodarone and thyroid autoimmunity was not confirmed. However, the authors speculate on the impact of amiodarone on pre-existing thyroid autoantibodies, because amiodarone usage per se did not initiate AIT [87,88].

#### 3.7.3. Lithium

Lithium therapy (LT) is the most common treatment in patients with bipolar disorders (BD). Lithium reduces manic episodes as well as the depressive side of the BD [88]. Despite the usefulness of LT in treating BD, it has some side effects that mainly affect the kidney and the thyroid [89]. LT may be the cause of hypothyroidism. Lithium concentrates in the thyroid gland and competitively reduces the iodine intake into the thyroid. This results in the decreased synthesis of thyroid hormones [90]. In other mechanisms, lithium inhibits the release of thyroid hormones and elevates the TSH level in patients. This results in a higher occurrence of goiter cases and hypothyroidism in patients during LT [90].

In studies, LT was found to increase the occurrence of TD and TSH titers in patients. Additionally, patients with LT were observed to have a greater volume of thyroid, and goiter could be found more often. By gender, the females in the study group were more likely to have thyroid dysfunction or a thyroid disorder compared to the males in the same group. Regarding AIT, no correlation was found between the LT and TPOAb or TgAb levels [91,92].

#### 3.7.4. Tyrosine Kinase Inhibitors

Tyrosine kinase inhibitors (TKIs) is a group of drugs used in the treatment of chronic myeloid leukemia (CML) with the positive Philadelphia chromosome. TKIs possess some adverse effects that may affect the thyroid. Hypothyroidism is one of the TKIs that influence the thyroid function. There are multiple pathways leading to hypothyroidism: the blockage of iodine intake, thyroid cell toxicity, leading to destructive thyroiditis, or the inhibition of thyroid peroxidase [93].

Rodia et al. [93] enrolled 69 CML-positive patients in the study. The study group was divided into three-subgroups representing different treatments, which included imatinib, nilotinib, and dasatinib. From all of the three subgroups, 21 patients met the criteria of HT. Out of all patients, seven were receiving imatinib, nine were receiving nilotinib, and five were receiving dasatinib. In this study. overt hypothyroidism was representative in two patients. Additionally, another two participants developed subclinical hypothyroidism. Both hypothyroidism groups were observed in HT patients [93].

### 3.8. Stress

The final discussed environmental factor that can be a potential trigger for autoimmune thyroiditis is stress. However, there have not been many studies that would clearly confirm or deny this theory. 

Some findings may suggest that acute, as well as chronic stress, may disturb the homeostasis of the human immune system and contribute to the development of autoimmunity. Disturbances in homeostasis may be a consequence of the hormonal imbalance caused by the activation of the sympathetic-adrenal system and the hypothalamic–pituitary–adrenal axis due to stressful situations. Catecholamines are released in excessive amounts, and there is an overproduction of glucocorticosteroids, including cortisol [94]. Furthermore, it has been shown that chronic stress is related to systemic inflammation, due to an increase in the level of pro-inflammatory cytokines, which may influence the development of autoimmune diseases such as AIT [94].

As well as this, in a study conducted by Hua Hong and Jeonghun Lee, the level of TSH was analyzed as a probable stress biomarker [95]. Normally TSH is regulated by fT3 and fT4 levels in a negative feedback loop. However, it was observed that cortisol, as well as cytokines levels, may also affect the serum concentration of TSH. Unfortunately, despite the potential of TSH as a reliable marker of stress, none of the stress questionnaires completed by the participants showed any association with TSH levels. The probable cause of this was the coexistence of too many confounding factors, such as uncontrollable environmental factors, personality variables, or emotional regulation strategies [95].

Another study found significantly higher levels of negative stress in AIT patients compared to controls, despite a very similar number of stressful life events in both groups. Moreover, this study showed a correlation between the number of stressful life situations and the level of anti-TPO antibodies, thus providing evidence of newly formed changes in immune pathways caused by stress [94].

### 3.9. Smoking

One of the considered environment factors, which can be correlated with AIT and thyroid function, is tabaco usage. Tobacco smoke contains numerous ingredients, such as nicotine, anatabine, carbon monoxide, and various carcinogenic substances such as free radicals, aldehydes, and aromatic hydrocarbons [96].

Some studies indicate a protective impact of smoking on the occurrence of HT. In a study presented by Strider et al., it was presented that the group of AIT patients who currently smoked cigarettes and did not show the presence of antibodies was larger (38%) than the group of smokers who presented a positive level of antibodies (25%, OR 0.69, 0.48–0.99) [97]. In the study designed by Belin et al., TPO-Ab were present in 11% of smokers vs. 18% of nonsmokers (OR 0·57, 0·48–0·67) [98]. These results were confirmed by other authors in Danish [99] and Teheran [100] populations. Mehran et al. also found that the association between antibodies and tobacco use was stronger for Tg-Ab than for TPO-Ab [101].

The mechanism of the protective effect of smoking on HT is likely related to the presence of alkaloids such as nicotine and antabine. Nicotine binds to nicotinic receptors, the expression of which also occurs on immune CD4+ T cells, dendritic cells, and macrophages. Through the α7-nicotine-acetylcholine receptor on Treg cells, nicotine can lead to the enhanced immune suppression of lymphocytes [101]. The other alkaloid, anatabine, unlike nicotine, is not addictive. It has a longer 8 h half-life. Anatabine shows effects on the immune system by reducing the incidence of Tg-Ab and by affecting macrophage activity through the inhibition of nitric oxide synthase and cyclooxygenase-2 [102].

In contrast to HT, smoking has been identified as a significant risk factor in Graves’ disease [97]. Moreover, it is emphasized that the risk is significantly higher in women than in men. It is worth emphasizing that even passive smoking is associated with an increased risk of ophthalmopathy: the incidence of ophthalmopathy among children with Graves’ hyperthyroidism is highest in countries with the highest prevalence of adolescent smoking [103].

### 3.10. Climate 

Climate change is causing impacts not only on our lives but also on our health. Some studies confirm the effect of environmental and weather conditions on autoimmune processes [104,105].

People living in areas with much lower temperatures and harsh climates, such as Siberia, show a much higher metabolism compared to groups of people living in warmer parts of the world. Thyroid hormones appear to play a key role in adaptation and increased thermogenesis. Exposure to harsh climatic conditions can lead to the induction of autoimmune processes [104].

Cepon et al. examined the population of Yakut (Sakha) of northeastern Siberia. Their study results were as follows: 22% of women and 6% of men had clinically elevated (>30 IU/mL) TPOAb, which was positively correlated with TSH (*p* < 0.01). Their study confirmed an increased susceptibility to AITD among women but also indicated that climate factors can influence the autoimmune thyroid process [105].

### 3.11. The Influence of Environmental and Genetic Factors on Immunological Processes in AIT

The etiopathogenesis of AIT is a combination of genetic, epigenetic, environmental, and immunological factors. The human leukocyte antigen (HLA) system provides a link between genetic and immunological factors in the development of AIT. The gene that is specifically associated with the development of AIT is HLA-DR3 [106]. However, there are numerous other immune-related genes that influence susceptibility to AIT. This group includes genes encoding cytotoxic T-lymphocyte antigen-4 (CTLA-4), protein tyrosine phosphatase non-receptor type 22 (PTPN22), CD40, and a variety of cytokines such as IL-1, IL-6, IL-10, IL-13, and TNF [107,108]. For illustration, a study conducted by Lacka et al. has shown the association between the interleukin *IL1* gene polymorphisms SNP-511 and SNP + 3953 and an increased risk of Hashimoto’s thyroiditis development [109].

In patients with AIT, inflammation occurs in the thyroid gland tissues, involving both B and T lymphocytes, especially Th1, Th2, and Th17 [107]. The development of Hashimoto’s disease is correlated with an immune response dependent mainly on Th1 lymphocytes, which is responsible for the mechanism of the cellular response and apoptosis of follicular cells in the thyroid gland [110]. The development of Th1 cells is dependent on IL-12 and IFN-γ, while Th1 lymphocytes produce the pro-inflammatory cytokines IL2, IFN-γ, TNF, and IL-1b, which leads to the activation of macrophages and cytotoxic effects [111]. 

Moreover, the immune response strongly correlates with environmental factors. Studies indicate that patients with *H. pylori* exhibit an increased Th1-related immune response, characterized by increased levels of IFN-γ, TNF, IL-1β, IL-6, IL-7, IL-8, and IL-10 and reduced amounts of IL-4, associated with Th2 lymphocytes. It has also been confirmed that *H. pylori* infection, especially in adults, causes an increase in IL-17 levels [112].

According to a study by Zhou et al., balancing vitamin D levels in patients with deficiency may have an impact on the normalization of C-reactive protein (CRP) levels and the improvement of the course of diseases related to the immune system [113]. 

Moreover, microelements that influence thyroid function are also involved in inflammatory processes. For example, selenium is necessary for the proper functioning of selenoproteins, which protect the thyroid tissue against free radicals released during oxidative stress. Moreover, it is involved in the process of differentiation and proliferation of T cells, determining the correct number of Th lymphocytes [114].

Smoking impacts both innate and acquired immunity. Tobacco smoke and its components affect immune mechanisms in various ways by increasing the intensity of the immune response or, in contrast, reducing it. The adaptive immune cells affected by smoking mainly include Th1, Th2, and Th17, which are also involved in AIT pathogenesis. Furthermore, CD4+, CD25+ regulatory T cells, CD8+ T cells, B cells, and memory cells are affected by smoking. The components of tabaco smoke influence immunoregulation processes by acting at the cellular and molecular levels. Innate immunity is altered by smoking, primarily through DCs, macrophages, and NK cells activity [115].

All environmental factors associated with autoimmune thyroiditis are listed in Table 5.

## 4. Conclusions

In conclusion, various environmental factors may influence the development of autoimmune processes in thyroid tissues.

Excessive iodine consumption should be highlighted as a risk factor. The correct intake of iodine in the daily diet may prevent the development of AIT.

Furthermore, vitamin D deficiency is an important but still unclear risk factor for AIT. More research should be conducted to reveal the relationship between vitamin D deficiency and the development of AIT. However, proper vitamin D intake appears to be a protective factor for the development of AIT.

Selenium deficiency may also contribute to the development of autoimmunity. Studies have shown that selenium supplementation can improve thyroid function in terms of AIT. However, more research on larger populations is necessary.

Research also confirms the relationship between viral and bacterial infections and the development of AIT, which may result from molecular mimicry. Studies have shown a relationship between EBV, PVB19, HHV-6A, SARS-CoV-2, and *Helicobacter pylori* infections with the etiopathogenesis of autoimmune thyroiditis. The effective prevention and treatment of infections may influence the occurrence and course of AIT.

Also, intestinal microflora disorders, increased intestinal wall permeability, and the activation of the immune system may be associated with the development of AIT. These processes indicate that maintaining proper intestinal microbiota and the appropriate use of probiotics and prebiotic substances may also affect the incidence and course of AIT.

The drugs used also have an impact on autoimmune processes. Many studies have shown an increased risk of AIT in patients undergoing IFN-α therapy, mainly HCV and HBV infections. The role of amiodarone and lithium therapy in the occurrence of AIT remains unclear and requires more research.

A psychological factor that may be related to the pathogenesis of AIT and its exacerbation is stress, but the research results remain unclear. In addition, more detailed interview methods should be developed to make research results on stressful events more accurate, which would increase the reliability of research results.

Smoking should be mentioned as a non-obvious factor protecting against the development of AIT, but a similar effect has not been confirmed in the case of Graves’ disease. It is worth emphasizing the role of alkaloids influencing autoimmune processes, the biological effects of which may in the future be used in the prevention and treatment of AIT.

Interestingly, the autoimmune process of the thyroid gland, which can be influenced by many different factors, can even be linked to climate change and weather factors. An increased predisposition to AIT appears to occur in populations living in harsh and cold parts of the world, which may shed new light on the occurrence of AIT in the context of ongoing climate change and global warming.

## Figures and Tables

**Figure 1 biomedicines-12-01788-f001:**
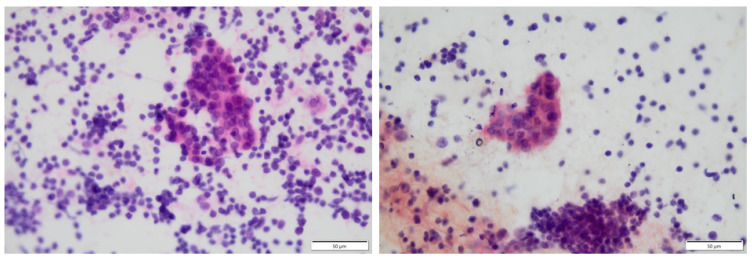
Thyroid image from fine needle aspiration (FNA) biopsy; Hürthle cells surrounded by massive lymphocytic infiltration.

**Figure 2 biomedicines-12-01788-f002:**
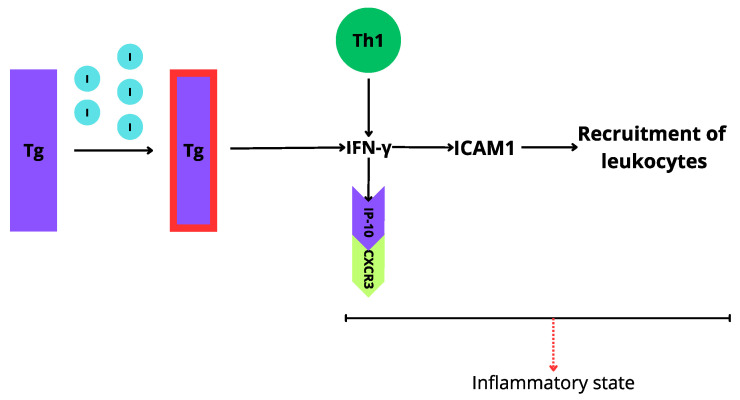
Iodine excess process leading to thyroid autoimmunity caused by inflammatory state. Tg-thyroglobulin, I—iodine excess, IFN-γ—interferon-γ, Th1—T helper cells 1, ICAM1—intercellular adhesion molecule-1, IP-10—(IFNγ)-induced protein 10, CXCR3—CXC chemokine receptor 3.

**Figure 3 biomedicines-12-01788-f003:**
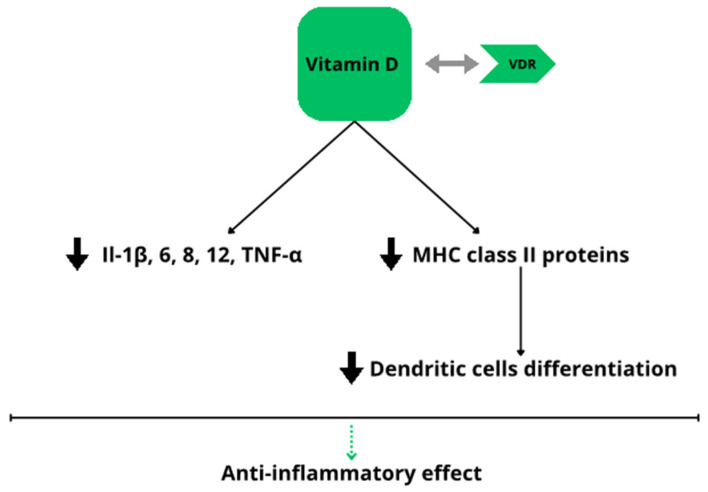
Vitamin D—anti-inflammatory effect, VDR—vitamin D receptor, IL—interleukin, TNF-α—tumor necrosis factor α.

**Figure 4 biomedicines-12-01788-f004:**
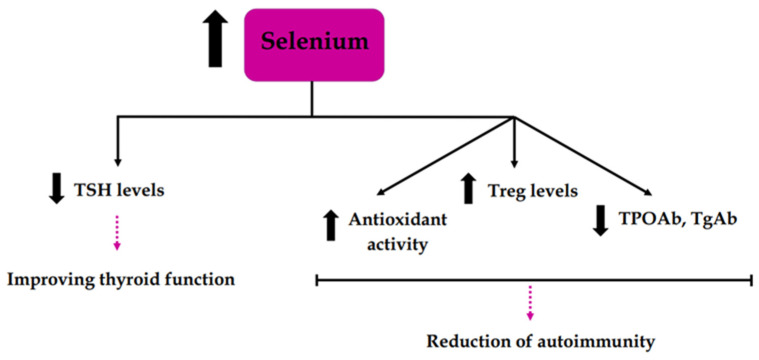
Effect of selenium supplementation on thyroid function and autoimmunity. TSH—thyroid stimulating hormone, Treg—T regulatory lymphocyte, TPOAb—thyroid peroxidase antibodies, TgAb—thyroglobulin antibodies.

**Figure 5 biomedicines-12-01788-f005:**
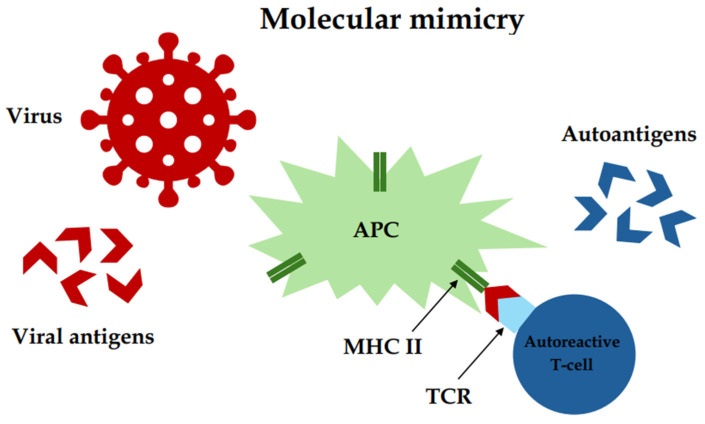
Molecular mimicry—a potential cause of autoimmunity due to the similarity between autoantigens and viral antigens. APC—antigen-presenting cell, MHC II—II class of major histocompatibility complex, TCR—T-cell receptor.

**Table 1 biomedicines-12-01788-t001:** Research results on the impact of iodine excess on AIT.

First Author	Patient Population (Nationality)	Total Patients	Control Group	Iodine Level	*p*	Iodine Excess Induces AIT
Lee et al. [20]	South Korean	439	No control group	UIC: 606.2 µg/L	0.021	-(hypothyroidism)
Yokomichi et al. [28]	Japanese	100,286	No control group	-	-	-(hypothyroidism)
Kasiyan et al. [24]	Ukrainian	168	68	-	-	+
Yu et al. [22]	Chinese	1159	182	MUI: 233.20 µg/L	<0.05	+
Hongyan et al. [26]	Chinese	160	60	MUI: iodine excess group: 212.69 µg/L iodine over-dose group: 302.51 µg/L	-	+
Palaniappan et al. [27]	Indian	86	43	UIE: 329.53 µg/L	<0.001	+
Novak et al. [25]	Slovenian	2889	1399	-	-	-(hypothyroidism)

UIC—Urinary iodine concentration, MUI—Median urinary iodine, UIE—Urinary iodine excretion.

**Table 2 biomedicines-12-01788-t002:** Research results on the impact of vitamin D deficiency on AIT.

First Author	Patient Population (Nationality)	Total Patients	Control Group	Mean Vitamin D Level in AIT Patients	*p*	Vitamin D Deficiency Induces AIT
Maciejewski et al. [34]	Polish	94	32	20.09 nmol/L	0.014	+
Siddiq et al. [31]	Pakistani	144	72	21.89 nmol/L	0.001	+
Chao et al. [33]	Chinese	5230	4889	15.81 ng/mL	0.014	+
Gašić et al. [35]	Serbian	156	48	20.23 ng/mL	<0.001	+
Ke et al. [47]	Chinese	175	63	45.77 nmol/L	<0.001	+
Rola et al. [41]	Polish	98	42	18.3 ng/mL	-	-
Anaraki et al. [42]	Iranian	65	32	-	-	-
Sönmezgöz et al. [39]	Turkish	136	68	16.85 ng/mL	<0.001	+
Xu et al. [36]	Chinese	394	200	40.4 nmol/L	<0.001	+
Kim et al. [37]	South Korean	776	407	92.1 nmol/L	>0.05	+
Ma et al. [48]	Chinese	210	70	31.00 nmol/L	<0.001	+
Yavuzer et al. [43]	Turkish	83	34	19.5 ng/mL	0.27	-
Gierach et al. [38]	Polish	370	125	27.01 ng/mL	<0.001	+
Cvek et al. [44]	Croatian	637	176	17.1 ng/mL	0.277	-
Evliyaoğlu et al. [40]	Turkish	169	79	16.67 ng/mL	0.001	+
Botelho et al. [45]	Brazilian	159	71	26.4 ng/mL	0.1917	-
Filipova et al. [46]	Slovakian	98	41	73 nmol/L ^a,b^ 55.7 nmol/L ^a,c^	0.8 0.9	-

^a^ AITD patients, ^b^ serum concentration of 25-hydroxyvitamin D in summer, ^c^ serum concentration of 25-hydroxyvitamin D in winter.

**Table 3 biomedicines-12-01788-t003:** Research results on the impact of selenium deficiency on AIT.

First Author	Patient Population (Nationality)	Total Patients	Control Group	Mean Se Level in AIT Patients	*p*	Selenium Deficiency Induces AIT
Wu et al. [51]	Chinese	1254	Unknown number ^d^	<80 μg/L	-	+
Cinemre et al. [52]	Turkish	82	42	148.90 ± 32.30 μg/L	0.002	+
Rostami et al. [53]	Iranian	99	50	0.87 ± 0.29 µmol/L	<0.001	+
Hu et al. [54]	Chinese	90	47	73.3 μg/L	0.493	- ^f^
Wang et al. [55]	Chinese	89	No control group	72.93 ± 43.242 ng/mL	<0.05 ^e^	- ^f^
Pirola et al. [56]	Italian	192	96	-	-	- ^f^
Manevska et al. [57]	North Macedonian	500	No control group	-	-	- ^f^
Szeliga et al. [50]	Polish	53	36	56.67 μg/L	*p* > 0.05	-

^d^ participants from the adequate-Se county (Ziyang); ^e^ after Se supplementation; ^f^ a positive impact of Se supplementation in AIT patients.

**Table 4 biomedicines-12-01788-t004:** Research results on the impact of viral infections on AIT.

First Author	Patient Population (Nationality)	Total Patients	Control Group	Viral Infections Induce AIT	Type of Virus
Al-Rammahi & Al-Khilkhali [60]	Iraqi	120	60	+	EBV
Assaad et al. [61]	Egyptian	120	60	+	EBV
Heidari & Jami [62]	Iranian	1132	480	+	PVB19
Seyyedi et al. [63]	Iranian	242	32	+	HHV-6A
Tesch et al. [59]	German	2,201,764	1,560,357	+	SARS-CoV-2
Weider et al. [68]	Norwegian	53	18	+	HHV-6A, Enteroviruses BPV19

**Table 5 biomedicines-12-01788-t005:** Environmental factors associated with autoimmune thyroiditis.

Environmental Factors Associated with Autoimmune Thyroiditis			
**Iodine excess**			
**Vitamin D deficiency**			
**Selenium deficiency**			
**Viral infections**			
EBV ^13^	PVB19 ^14^	HHV-6A ^15^	SARS-CoV-2 ^16^
**Bacterial infections**			
** *Helicobacter pylori* **			
**Microbiome disruption**			
**Medications**			
**Interferon-alpha**	Amiodarone	Lithium	Tyrosine kinase inhibitors
**Stress**			
**Smoking**			
**Climate**			

^13^—Epstein–Barr Virus (EBV); ^14^—Human parvovirus B19 (PVB19); ^15^—Human herpesvirus 6A (HHV-6A); ^16^—Severe acute respiratory syndrome coronavirus 2 (SARS-CoV-2).

## Data Availability

The data presented in this study are available on request from the corresponding author.

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
