# Peer review of "The Impact of Environmental Factors on the Development of Autoimmune Thyroiditis—Review"

_biomedicines, 2024, doi:10.3390/biomedicines12081788_

Round 1
Reviewer 1 Report
Comments and Suggestions for Authors
Thank you for the possibility to review the manuscript titled: «The impact of environmental factors on the development of autoimmune thyroiditis – Review». The review is well written and easy to read. The reference list includes over 90 articles on the topic and cites most of the available important literature. There are no major objections. However, I would recommend several minor recommendations:
-Please include some data about thyroiditis complications. Decrease thyroid function is not the only concern. Chronic thyroiditis can lead to malignancy, hemorrhage, extrathyroid complications, which should be mentioned in the article. A list of potential complications: goiter, heart problems, mental health issues, sexual and reproductive dysfunction, poor pregnancy outcomes, myxedema.
-There are multiple other viruses that are linked to thyroiditis and one of the major players are enteroviruses. I’ve provided to potential links for this relationship.
1) Weider T, Genoni A, Broccolo F, Paulsen TH, Dahl-Jørgensen K, Toniolo A, Hammerstad SS. High Prevalence of Common Human Viruses in Thyroid Tissue. Front Endocrinol (Lausanne). 2022 Jul 14;13:938633. doi: 10.3389/fendo.2022.938633. PMID: 35909527; PMCID: PMC9333159.
2) Desailloud, R., Hober, D. Viruses and thyroiditis: an update. Virol J 6, 5 (2009). https://doi.org/10.1186/1743-422X-6-5
Please take into consideration the recommendation in the spirit of improving the quality of the submission.
Reviewer 2 Report
Comments and Suggestions for Authors
The review paper titled "The impact of environmental factors on the development of autoimmune thyroiditis - Review" is carefully read and reviewed. TAuthors aimed to collect and analyze data related to the environmental factors influencing the development of autoimmune thyroiditis. The findings indicate that iodine intake, vitamin D deficiency, selenium deficiency, viral infections (such as those caused by Epstein-Barr Virus [EBV], Human parvovirus B19 [PVB19], Human herpesvirus 6A [HHV-6A], and Severe acute respiratory syndrome coronavirus 2 [SARS-CoV-2]), bacterial infections (such as those caused by Helicobacter pylori), microbiome disruption, medications (including interferon-alpha and tyrosine kinase inhibitors), as well as stress, climate, and smoking, can influence the risk of developing autoimmune thyroiditis. The paper is organized well enough. However, I recommend authors emphasizing the role of inflammation as a link between autoimmune thyroiditis and environmental factors. Because, autoimmune thyroiditis is characterized with elevation in inflammatory markers, such as uric acid to HDL ratio, C-reactive protein, DeRitis ratio and hemogram derived inflammatory markers (i.e. neutrophil to lymphocyte ratio). Besides, vitamin D deficiency is linked with chronic inflammation as reported in type 2 diabetes mellitus and sarcopenia. Smoking, selenium deficiency, virus infections and H.pylorii infections are also associated with increased inflammatory burden. Hence, the review would benefit from discussing the role of inflammation in interaction between environmental factors and thyroiditis.
Round 2
Reviewer 2 Report
Comments and Suggestions for Authors
The revisions are satisfactory. all of the issues I raised were addressed appropriately.